# Personalised analytics for rare disease diagnostics

Denise Anderson [1*], Gareth Baynam [2,3,4], Jenefer M. Blackwell [1] & Timo Lassmann [1*]

Whole genome and exome sequencing is a standard tool for the diagnosis of patients suffering from rare and other genetic disorders. The interpretation of the tens of thousands of variants returned from such tests remains a major challenge. Here we focus on the problem of prioritising variants with respect to the observed disease phenotype. We hypothesise that linking patterns of gene expression across multiple tissues to the phenotypes will aid in discovering disease causing variants. To test this, we construct classifiers that learn associations between tissue-specific gene expression and disease phenotypes. We find that using Genotype-Tissue Expression project (GTEx) expression data in conjunction with disease agnostic variant prioritisation methods (CADD or MetaSVM) results in consistent improvements in classification accuracy. Our method represents a previously overlooked avenue of utilising existing expression data for clinical diagnostics, and also opens the door to use of other functional genomic data sets in the same manner.

[1] Telethon Kids Institute, The University of Western Australia, PO Box 855West Perth, WA 6872, Australia. [2] Office of Population Health Genomics, Department of Health, PO Box 8172 Perth Business Centre, Perth, WA 6849, Australia. [3] Genetic Services of Western Australia, King Edward Memorial Hospital, PO Box 134 Subiaco, WA 6904, Australia. [4] Western Australian Register of Developmental Anomalies (WARDA), King Edward Memorial Hospital, PO Box 134 Subiaco, WA 6904, Australia. *email: Denise.Anderson@telethonkids.org.au; Timo.Lassmann@telethonkids.org.au

Clinical genome and exome sequencing has become widely adopted for diagnosis of genetic diseases since becoming commercially available in 2011[1]. Despite considerable efforts in this area, the majority of patients remain undiagnosed[2,3]. Whole genome sequencing (WGS) has many benefits over whole exome sequencing (WES) due to coverage of the entire genome versus the protein-coding genome. Entire genome coverage allows reliable detection of copy number variation and of variants in non-coding regions, both of which have been associated with disease[4]. In addition, WGS does not rely on capture technologies and hence offers better coverage in protein coding and GC rich regions compared with WES[5]. These advantages have seen WGS adopted as the default choice for genetic diagnoses in the United Kingdom[6], but regardless of the technology used the problem of discovering the causative variants remains the same.

When calling variants from WES, 60,000–100,000 variants are found on average[7]. The vast majority of these variants are benign and unrelated to the patient's disease. The challenge is therefore to discover the few variants which are likely to cause the disease for further investigations. This prioritisation task is achieved through a series of filtering steps. Common steps include, filtering by allele frequency in large sequencing cohorts and by the predicted impact of the variant on the protein sequence[8]. After filtering, the remaining variants are ranked by consulting disease-variant (e.g. ClinVar[9]) and disease-gene databases (e.g. OMIM[10]) to determine whether any variants have been previously reported to be associated with the disease under study or a similar disease. Further refinements to the list of candidate variants can be achieved through use of in-silico variant prioritisation tools[11,12].

To predict the pathogenicity or deleteriousness of variants, variant prioritisation tools use a number of features. They can be grouped into three main categories: (1) tools based on genome conservation, (2) tools predicting the effect of variants on protein function and (3) meta prediction tools combining multiple predictions. Currently, it is common practice to train and assess these methods based on the complete set of known disease causing variants. However, we recently demonstrated that the performance of these tools, including the performance of the best methods, varies considerably when applied to variants causing different diseases[13]. This result suggests that developing classifiers for variants in specific disease groups can improve prediction accuracy.

Variants that affect a cell's core set of specific genes can disrupt normal protein function, leading to pathogenic cell types and ultimately disease. Hence, if we are trying to identify causal variants for a patient with a neurological phenotype we should prioritise variants in neuronal genes higher than other variants. While we currently do not have a complete compendium of cell-type-specific gene expression across all conditions (e.g. populations, temporal, gender specific etc.), recent large scale consortia projects have amassed a considerable amount of data that could be used in this context[14,15]. Furthermore, we anticipate that projects including the Human Cell Atlas[16] will continue to provide data, further increasing our understanding of cell-type-specific gene expression and regulation.

Figure 1 illustrates the rationale behind the use of tissue and cell-specific gene expression for a disease phenotype associated with the brain. In silico predictions of pathogenicity from variant prioritisation tools are not phenotype aware, so there is an opportunity to improve these predictions. In our example, some pathogenic variants are predicted to be benign (P1, P5 and P10) and some benign variants are predicted to be pathogenic (B3, B8 and B11). If instead we train classifiers in a phenotype aware manner, we expect there to be associations between pathogenic variants and the expression pattern of the gene harbouring the variant. In our illustrative example, we see that pathogenic variants are associated with gene expression in brain tissue and neuron cells. This association becomes apparent because we have reduced the training set to pathogenic variants within genes that are associated with the brain-specific phenotype. In this manner we expect expression data to improve causative variant classification when combined with disease-specific classifiers.

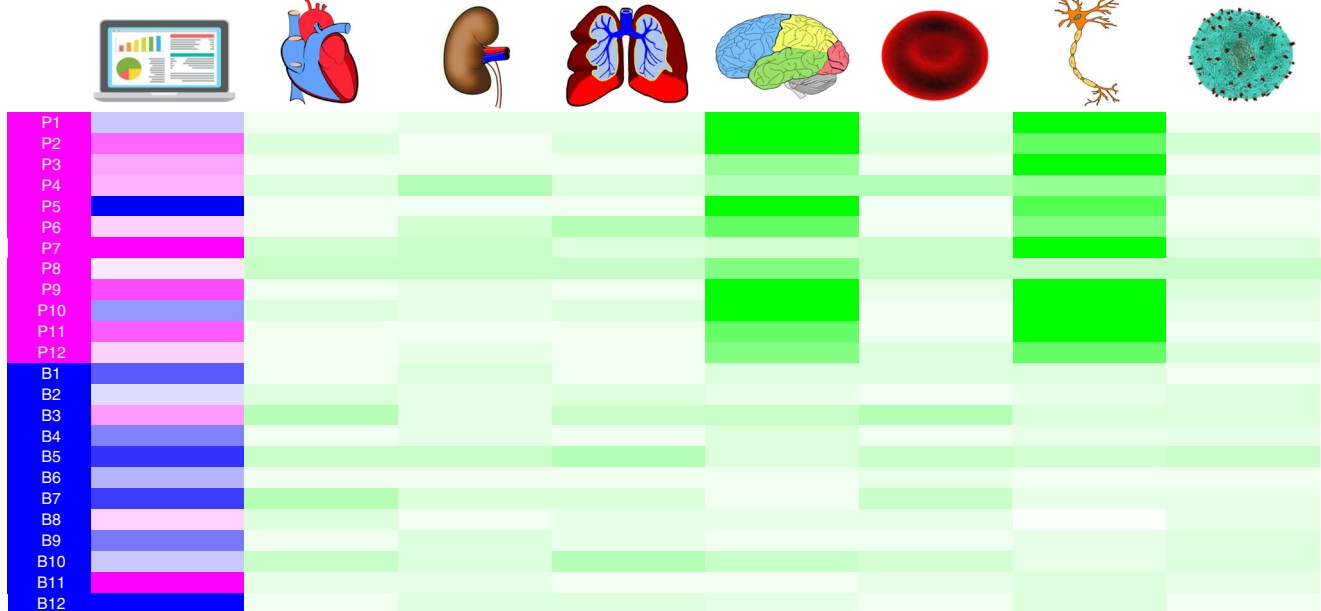

**Fig. 1** Rationale for use of tissue and cell-specific gene expression for prioritisation of variants associated with a brain disease phenotype. Pathogenic variants are coloured magenta and benign variants are coloured blue (first column). The second column shows in silico predictions of variant pathogenicity, where increasing magenta intensity indicates stronger probability of pathogenicity and increasing blue intensity indicates stronger probability of being benign. The remaining columns in order represent heart tissue, kidney tissue, lung tissue, brain tissue, red blood cells, neurons and T cells. The green colour scale represents gene expression, where increasing colour intensity indicates higher expression values.

Here we construct random forest classifiers for phenotypic abnormality terms found in the Human Phenotype Ontology (HPO)[17]. The classifiers include variant prioritisation scores in conjunction with tissue or cell-specific gene expression from the Genotype-Tissue Expression project (GTEx)[15] or the Functional ANnoTation Of the Mammalian genome consortium (FANTOM)[14,18]. We assess whether inclusion of gene expression leads to improvements in variant classification compared with using established variant prioritisation scores on their own. We call our method VARiant Prioritisation by Phenotype (VARPP) and test 1879 disease phenotypes.

## Results

**Performance of VARPP.** We assessed the performance of VARPP using auPRC and PP100 (as described in "Methods") against use of the variant prioritisation tools alone (Table 1, Supplementary Data 1). Performance gains are greater for VARPP using GTEx tissue expression or specificity, when compared with VARPP using FANTOM5 cell expression or specificity. In addition, VARPP classifiers including CADD outperform the respective VARPP classifiers including MetaSVM in terms of auPRC. The auPRC quantifies the ability of VARPP to correctly identify all pathogenic and benign variants, and under this scenario CADD is the superior choice when compared with MetaSVM. The opposite is true for PP100, where VARPP classifiers including MetaSVM outperform the respective VARPP classifiers including CADD. In a clinical diagnostic setting, the PP100 is the more useful measure, as it describes the ability to enrich for pathogenic variants within the top 100 rankings.

VARPP using GTEx expression data outperforms CADD and MetaSVM alone for most HPO terms when assessed using the PP100, but for the auPRC VARPP only outperformed CADD not MetaSVM (Table 1, Supplementary Data 1, Supplementary Fig. 1). For the auPRC we see an improvement across 1314 (70%) HPO terms for VARPP including CADD, versus using CADD alone. VARPP including MetaSVM performs worse for 1158 (62%) terms when compared with using MetaSVM alone. Though VARPP including MetaSVM did not show improvement for the majority of HPO terms based on the auPRC, it did based on the PP100 (1474 [78%] of terms improve), as did VARPP using CADD (1030 [55%] of terms improve).

When considering specificity rather than magnitude of expression, we see similar performance for VARPP including CADD, whereas VARPP including MetaSVM shows performance gains (Table 1, Supplementary Data 1, Fig. 2, Supplementary Fig. 2). We see improvement in the auPRC across 1251 (67%) terms for VARPP including MetaSVM, versus using MetaSVM alone. VARPP using MetaSVM also performs very well for the PP100 (1645 [88%] of HPO terms improve). For VARPP incorporating CADD we see very similar performance for the auPRC (1306 [70%] of HPO terms improve) and the PP100 (1057 [56%] of HPO terms improve) to that seen when using expression. We further note that improvements are consistent for VARPP including MetaSVM if instead we use the proportion of true pathogenic variants in the top 50 or 200 predictions of pathogenicity (PP50 = 1664 [89%] of HPO terms improve; PP200 = 1467 [78%] of HPO terms improve). This is also the case for VARPP including CADD for the PP200 (1198 [64%] of HPO terms improve), but there are fewer improvements for the PP50 (776 [41%] of HPO terms improve) due to HPO terms where true pathogenic variants in the top 50 have CADD scores close to 1 (i.e. there is little opportunity for

**Table 1 Performance of VARPP versus use of variant prioritisation tools alone.**

| VARPP | auPRC n (%)[a] | Difference (CI) | t (df) | P value | PP100 n (%)[b] | Difference (CI) | t (df) | P value |
|---|---|---|---|---|---|---|---|---|
| GTEx expression + CADD | 1314 (70%) | 0.042 (0.038 to 0.046) | 21.6 (1878) | <0.001 | 1030 (55%) | 0.051 (0.044 to 0.057) | 15.0 (1878) | <0.001 |
| GTEx expression + MetaSVM | 721 (38%) | −0.005 (−0.001 to −0.009) | −2.7 (1878) | 0.006 | 1474 (78%) | 0.062 (0.058 to 0.067) | 25.7 (1878) | <0.001 |
| GTEx specificity + CADD | 1306 (70%) | 0.042 (0.038 to 0.046) | 21.4 (1878) | <0.001 | 1057 (56%) | 0.051 (0.045 to 0.057) | 16.7 (1878) | <0.001 |
| GTEx specificity + MetaSVM | 1251 (67%) | 0.031 (0.027 to 0.036) | 14.6 (1878) | <0.001 | 1645 (88%) | 0.096 (0.092 to 0.100) | 43.7 (1878) | <0.001 |
| FANTOM5 expression + CADD | 705 (38%) | −0.013 (−0.009 to −0.016) | −6.7 (1878) | <0.001 | 797 (42%) | 0.011 (0.005 to 0.017) | 3.7 (1878) | 0.0002 |
| FANTOM5 expression + MetaSVM | 189 (10%) | −0.071 (−0.067 to −0.075) | −34.8 (1878) | <0.001 | 1127 (60%) | 0.015 (0.010 to 0.020) | 6.4 (1878) | <0.001 |
| FANTOM5 specificity + CADD | 581 (31%) | −0.026 (−0.022 to −0.030) | −13.6 (1878) | <0.001 | 637 (34%) | −0.015 (−0.009 to −0.021) | −4.7 (1878) | <0.001 |
| FANTOM5 specificity + MetaSVM | 264 (14%) | −0.058 (−0.053 to −0.063) | −24.1 (1878) | <0.001 | 1286 (68%) | 0.027 (0.021 to 0.032) | 9.7 (1878) | <0.001 |

[a]Number and percentage of HPO terms where VARPP performed better for the auPRC than the variant prioritisation tool alone. auPRC mean difference, 95% confidence intervals (CI), t statistic, degrees of freedom (df) and P value are from a Student's paired t test
[b]Number and percentage of HPO terms where VARPP performed better for the PP100 than the variant prioritisation tool alone. PP100 mean difference, 95% confidence intervals (CI), t statistic, degrees of freedom (df) and P value are from a Student's paired t test

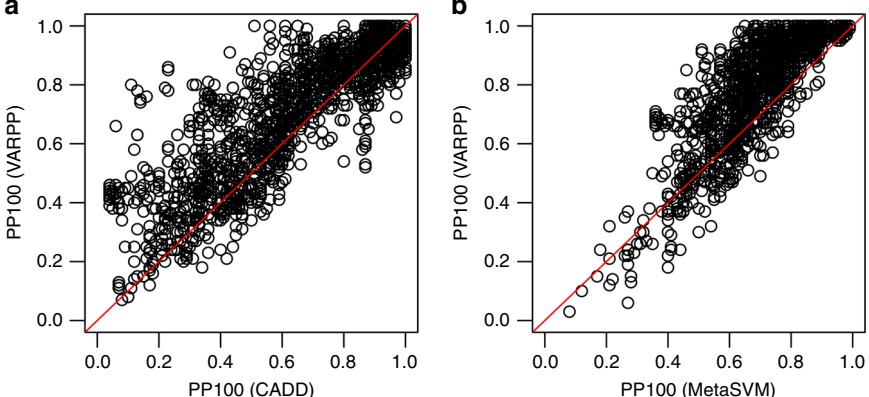

**Fig. 2** Performance of VARPP classifiers across 1879 HPO Phenotypic Abnormality terms. **a** Agreement scatter plot comparing the PP100 for VARPP including CADD + GTEx specificity (y axis) versus the PP100 for CADD scores alone (x axis). **b** Agreement scatter plot comparing the PP100 for VARPP including MetaSVM + GTEx specificity (y axis) versus the PP100 for MetaSVM scores alone (x axis). The red line is the line of identity.

VARPP to improve on predictions of variant pathogenicity in these scenarios).

Performance for both FANTOM5 expression and to a lesser extent specificity (Table 1, Supplementary Data 1, Supplementary Figs. 3 and 4), preclude use of these data sets in VARPP. We believe this is due to the small sample sizes assayed for each cell type and anticipate that larger data sets (such as those currently being generated by the Human Cell Atlas) will be required. In summary, we find GTEx expression data superior to FANTOM5 expression data when prioritising variants for the disease phenotypes considered here. In addition, we find specificity of gene expression to be preferred over magnitude of expression. Therefore, results henceforth will focus on VARPP using GTEx specificity. In practice we would not restrict our use of VARPP to a particular data set, instead we would use VARPP in combination with the data sets that show the best performance for the particular phenotype(s) being investigated and we provide Supplementary Data 1 for this purpose.

**Performance of VARPP by disease group.** To determine whether performance differs by distinct disease groups, we plotted results for each HPO term grouped by the broad disease category the term belongs to (Fig. 3, Supplementary Figs. 5–8). The auPRC for VARPP including CADD, versus use of CADD alone, shows improvement (>0.2) for many of the terms associated with Abnormality of the integument, Abnormality of the cardiovascular system, Abnormality of blood and blood-forming tissues and Abnormality of metabolism/homoeostasis (Supplementary Fig. 5). For VARPP including MetaSVM, improvement is most evident for Abnormality of metabolism/homoeostasis (Supplementary Fig. 6). When considering the PP100, VARPP including CADD showed improvement (>0.25) over use of CADD alone, for many of the terms associated with Abnormality of the nervous system, Abnormality of the skeletal system, Abnormality of the integument, Abnormality of metabolism/homoeostasis, Abnormality of the digestive system and Abnormality of limbs (Fig. 3, Supplementary Fig. 7). VARPP including MetaSVM showed improvement for most HPO terms (88%), hence consistent improvement is seen across many of the disease groups (Fig. 3, Supplementary Fig. 8). Therefore, improvement is achieved across a broad range of disease phenotypes, particularly for the PP100. These results will be useful for determining the best variant prioritisation method to use for particular HPO terms.

**Random forest variable importance.** Random forest classifiers return variable importances that quantify the importance of each predictor in predicting the outcome. This allows us to determine which GTEx tissues are most associated with each disease phenotype (Supplementary Data 2). We examined variable importance across the tissues for the top 30 improved HPO terms (Supplementary Figs. 9–12). The top improved terms (auPRC) for VARPP including CADD revealed several biological associations (Supplementary Fig. 9). In particular, the two most important tissues for restrictive cardiomyopathy and syncope (fainting) are heart (atrial appendage) and heart (left ventricle). Restrictive cardiomyopathy is a disease of the heart muscle and syncope is caused by a drop in heart rate and blood pressure. Fibroblasts derived from skin samples were associated with thin skin, dermal atrophy, abnormality of the sclera, blue sclerae, dilatation of the ascending aorta and aortic dilatation. Fibroblasts are found in connective tissue throughout the body including both the sclera and aorta. The most important tissue for abnormality of iron

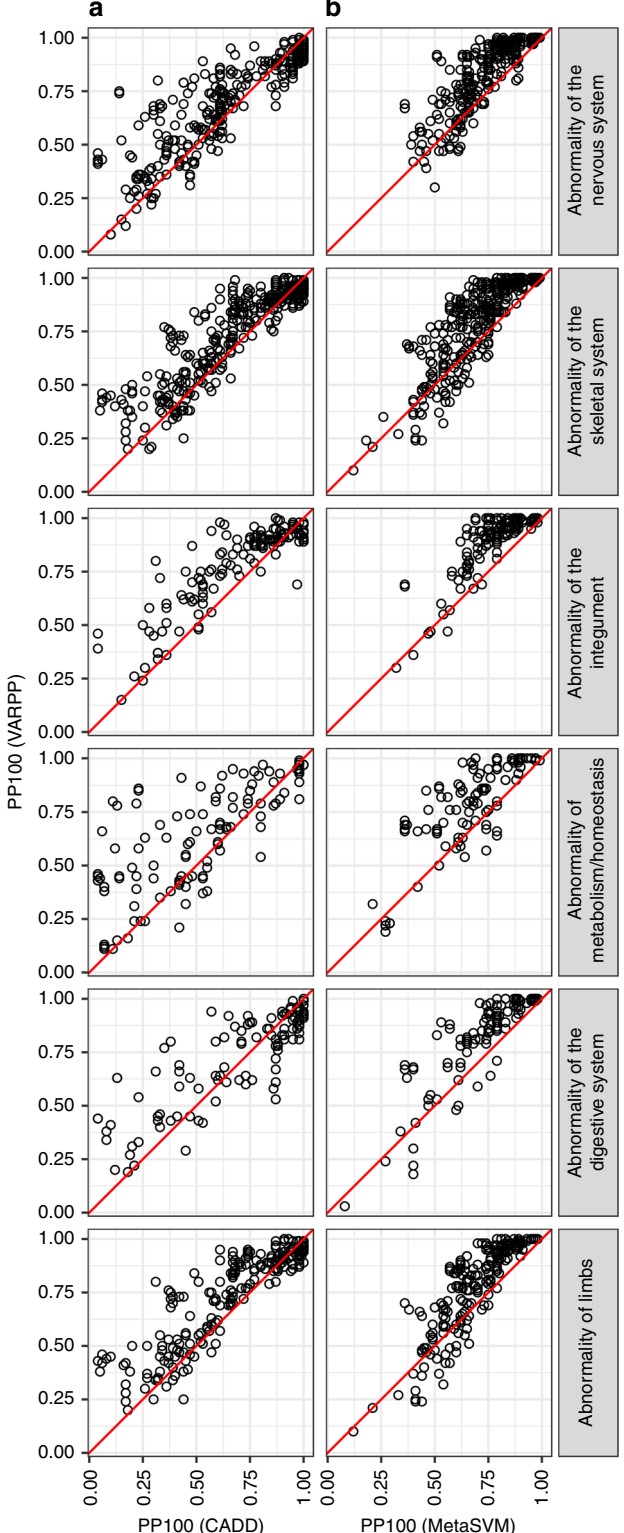

**Fig. 3** Performance of VARPP classifiers by disease group. **a** Agreement scatter plots comparing the PP100 for VARPP including CADD + GTEx specificity (*y* axis) versus the PP100 for CADD scores alone (*x* axis). **b** Agreement scatter plots comparing the PP100 for VARPP including MetaSVM + GTEx specificity (*y* axis) versus the PP100 for MetaSVM scores alone (*x* axis). The red line is the line of identity.

homoeostasis is whole blood and erythropoiesis (red blood cell production) is the greatest consumer of iron. Moreover, one of the possible symptoms of food intolerance is skin rash and skin is seen to be the most important tissue for this term. Lactic acidosis can be caused by heart disease due to reduced blood and oxygen flow, and the most important tissues for this term are heart (left ventricle) and artery (tibial). Veins and arteries have a similar anatomical structure consisting of three layers and we find that venous abnormality is most associated with artery (tibial). Cyanosis is visible as a bluish discolouration of the extremities and mucous membranes due to lack of oxygen in the blood. Acrocyanosis is the same condition but is specific to the extremities. For both terms we see associations with adipose (subcutaneous) and artery (coronary). Adipose (subcutaneous) is a component of the skin where the bluish discolouration is observed and artery (coronary) is involved in blood circulation. In addition to these associations, cyanosis is also associated with oesophagus (gastroesophageal junction) which is lined with a mucous membrane. When considering the top improved terms based on the top 100 (Supplementary Fig. 11) we see some of the aforementioned associations (blue sclera, dilatation of the ascending aorta and lactic acidosis) and also a number of relevant terms associated with whole blood (hypochromic anaemia, thrombophlebitis, microcytic anaemia, abnormality of transition element cation homoeostasis, abnormality of iron homoeostasis, leukocytosis).

Variable importances from VARPP including MetaSVM (Supplementary Figs. 10 and 12) show weaker associations than VARPP including CADD. This suggests that improvements in performance are due to associations across multiple tissues.

We also examined the HPO term assigned the highest variable importance for each tissue, which revealed plausible biological associations in adrenal gland, brain (basal ganglia), brain (other), heart (atrial appendage), heart (left ventricle), kidney, liver, muscle, pituitary, salivary gland, skin, spleen, vagina and whole blood (Supplementary Table 1). In summary, VARPP reveals associations between disease phenotypes and biologically relevant tissues. This is an encouraging result for our approach, and we plan to widen our scope to other functional genomic data sets (and combinations of these data sets) in future work.

**Performance in simulated disease exomes**. We randomly spiked ClinVar pathogenic variants into healthy exomes sequenced as part of the 1000 Genomes Project to assess how our approach performs when attempting to identify a single causal variant, amongst benign individual variation. Most spike-in variants were ranked higher than or equal to both CADD (2043/3079 = 66%) and MetaSVM (1308/2046 = 64%) alone when using rankings from VARPP (Supplementary Fig. 13, Supplementary Data 3). Practically, entire sets of ranked variants will not be examined as resource limitations will restrict the number of variants that can be functionally validated. Therefore, we investigated to what extent the spike-in variants appeared within the top 50 ranked variants. 66% (2044/3079) of the spike-in variants were ranked in the top 50 for VARPP including CADD, versus 57% (1777/3101) when using CADD alone. A similar percentage of spike-in variants were ranked in the top 50 for VARPP including MetaSVM (1516/2046 = 74%) compared with MetaSVM alone (1545/2058 = 75%). Importantly though, if we restrict results to variants that have a rank of 50 or lower with either VARPP or the variant prioritisation tool, we find that 74% (1664/2236) of spike-in variants were ranked higher than or equal to CADD and 70% (1141/1624) of variants were ranked higher than or equal to MetaSVM. These results show great utility for the use of VARPP in clinical settings where the focus will be on a small set of top candidate variants. We envisage that

the VARPP ranking can be used alongside rankings from other recommended variant prioritisation tools[13], as per the American College of Medical Genetics and Genomics recommendation to consult multiple in silico prediction tools[19].

## Discussion

We find that use of specificity of tissue gene expression in disease-specific classifiers does improve variant pathogenicity prediction (Table 1, Fig. 2, Supplementary Fig. 2). Use of magnitude of gene expression in tissues also results in performance gains (Table 1, Supplementary Fig. 1), but not to the extent seen for specificity of expression. Neither cell-type-specific gene expression or specificity showed consistent utility for improving predictions of variant pathogenicity (Table 1, Supplementary Figs. 3 and 4). This is contrary to what we expected prior to conducting this study. We anticipated that cell-specific data would be of more use than tissue-specific data because the finer resolution should better characterise genome function. The most likely reason is that FANTOM5 assayed much smaller sample sizes per cell type (typically no more than three) in comparison to GTEx tissues (typically hundreds). We averaged expression across each tissue or cell-type group and this estimate will be more accurate in GTEx than in FANTOM5 due to the larger sample sizes. This additional noise in the FANTOM5 data will make it more difficult to detect associations with pathogenic variants. Another explanation for these results could be that expression at the whole tissue level is more biologically relevant to diseases than the expression in individual cell types. In many instances we have no knowledge about the mechanisms that lead to a particular disease phenotype, so we expect varying success across different functional genomic data sets.

Specificity of tissue gene expression was preferred to magnitude of expression for VARPP including MetaSVM, but performance was similar for either measure when considering VARPP including CADD (Table 1, Fig. 2 and Supplementary Figs. 1 and 2). Neither CADD nor MetaSVM (or the nine existing variant prioritisation scores included in the MetaSVM ensemble classifier) used GTEx expression data in their classifiers so this result is not due to overfitting or collinearity. Rather, we believe this is due to CADD and MetaSVM scores being based on different methodologies and hence each score interacts differently with the functional genomics data sets used in the classifiers. Practically, we expect specificity to be the more biologically meaningful measure for our approach and will adopt this in future work.

Further to the performance of VARPP, we acknowledge that our strategy of selecting variants with minor allele frequencies of at least 0.01 as our benign set of variants could bias performance estimates of VARPP, because benign variants with minor allele frequencies <0.01 may differ in their distribution of CADD and MetaSVM scores. Whether this could lead to over or underestimation of VARPPs performance is impossible to assess due to lack of reliable annotation for rare benign variants. Any bias would be minimal though, due to our large sample of benign variants, and our strategy is preferable to avoid inclusion of rare functional variants. Furthermore, the confidence intervals presented in Table 1 may be too narrow given the overlap of genes and variants across HPO terms. Finally, the higher proportion of missing scores for MetaSVM (15.2% across all pathogenic and benign variants) in contrast to CADD (0.03%), means comparisons across VARPP classifiers is imperfect. This also needs to be considered when using VARPP to predict pathogenicity of an individual patient's variants, as less will be scored for VARPP classifiers including MetaSVM compared with VARPP classifiers including CADD.

Relevant tissues were associated with the most improved terms for VARPP including CADD (Supplementary Figs. 9 and 11), but

less so for VARPP including MetaSVM (Supplementary Figs. 10 and 12). When considering the HPO term associated with the maximum variable importance for each tissue, the terms are often biologically relevant (Supplementary Table 1). We used the permutation importance measure and this is not ideal for our purposes given the imbalance between pathogenic and benign variants as it is based on overall prediction accuracy. In our case, even if the tissue is quite important for prediction of pathogenic variants, the variable importance will be dominated by benign variants due to the imbalance. Hence, the magnitude of variable importances across the tissues was very small (maximum = 0.006 for VARPP including CADD; maximum = 0.005 for VARPP including MetaSVM). A more suitable measure of variable importance would be based on the auPRC rather than accuracy, but development of such a measure was beyond the scope of this study. As expected, the maximum variable importances for CADD and MetaSVM were larger (0.13 for CADD; 0.17 for MetaSVM), given that these scores were developed specifically for the task of separating pathogenic and benign variants. Despite the aforementioned limitations, we believe the importance measure used here is capable of highlighting a tissue's importance relative to other tissues.

VARPP performs well when attempting to identify a single pathogenic variant in an individual exome (Supplementary Fig. 13). Our aim in developing VARPP was to improve ranking of causal variants for patients and these results support the utility of our approach. We will improve implementation of VARPP including offering actual impurity reduction (AIR) variable importance[20] and variable importance $p$ values[21,22], as additional sources of evidence for determining variable importance. We are also currently working on improving run times for VARPP and on separate functions for training and prediction, for the convenience of users who would like to use the same random forest repeatedly for predictions on new data.

We have only considered gene expression here, but our approach can use any type of functional genomics data provided that many conditions, tissues and/or time points are sampled. We plan to expand our approach to the non-coding regions of the genome, as this will be essential to fully elucidate disease mechanisms. Given the encouraging results for gene expression, we envisage additional gains can be made by considering data sets that assay regulators of gene expression such as promoters, enhancers and non-coding RNA to name a few. Many of these data sets will also allow for more precise spatial mapping of functional data to variants, compared with the mapping of gene expression performed in our study. Including such data sets simultaneously in VARPP is also desirable but further work is required to determine whether batch effects (due to differing technologies) would dominate biological signals. Initiatives like the Human Cell Atlas are bringing us closer to understanding cellular function and how this function is affected by disease. Development of novel in silico approaches work hand in hand with these large data sets and will continue to be essential for understanding disease. These same approaches show translational promise in helping to address critical clinical analysis needs for genetic and rare diseases.

## Methods

**Integration of phenotype with annotated variants**. We previously described in detail each component of an automated pipeline to integrate phenotypes with annotated variants[13]. Therefore, we only briefly describe these aforementioned components (human phenotype ontology, linking disease phenotypes to genes using Phenolyzer, pathogenic variants) and focus on the new components.

**Human phenotype ontology**. We retrieved 12,461 HPO Phenotypic Abnormality terms and used package ontologyIndex[23] within R 3.4.3[24] to read in the obo file downloaded from https://hpo.jax.org/app/download/ontology on 23 June 2017.

**Linking disease phenotypes to genes using Phenolyzer**. Retrieval of disease genes associated with the HPO terms, was performed using Phenolyzer[25]. We used the command line version (https://github.com/WGLab/phenolyzer), with options specified to generate the same result as the Phenolyzer web server with default settings (-p -ph -logistic -addon DB_DISGENET_GENE_DISEASE_-SCORE,DB_GAD_GENE_DISEASE_SCORE -addon_weight 0.25). We only considered genes with a known disease association (seed genes in Phenolyzer). We did not apply a threshold to genes selected for each HPO term based on the Phenolyzer confidence score.

**Pathogenic variants**. Variants in the seed genes returned by Phenolyzer were annotated using dbNSFP version 3.4a (release 12 March 2017)[26,27]. We selected ClinVar[9] pathogenic variants and used the converted rank scores for CADD[12] and MetaSVM[11]. We removed ClinVar pathogenic variants with minor allele frequencies greater than or equal to 0.01 in the following cohorts: (1) 1000 Genomes[28], (2) UK10K_COHORT_TWINSUK[29], (3) UK10K_COHORT_ALSPAC[29], (4) NHLBI GO Exome Sequencing Project[30], and (5) Exome Aggregation Consortium[31] as they are unlikely to be causal variants. After selecting pathogenic variants within the seed genes for each HPO term, we excluded terms having fewer than 25 genes. In total we produced 1879 data sets, one for each HPO term, with a median of 1073 pathogenic variants (Supplementary Data 4).

**Benign variants**. We used package org.Hs.eg.db[32] to retrieve all gene symbols across the human genome. Variants within these genes were annotated using dbNSFP version 3.4a (release 12 March 2017). We defined benign variants as those not present in ClinVar and common in at least one population (MAF ≥ 0.01 in the same populations as above). We excluded benign variants present in the 2916 genes containing pathogenic variants. After filtering, there were 55,523 benign variants across 12,313 genes (Supplementary Data 5). When considering both the pathogenic variants (described above) and the benign variants, MetaSVM had a higher rate of unscored variants (15.2%) than CADD (0.03%), but we decided not to subset the data to those variants with scores from both tools to include the maximum number of variants.

**Gene expression data**. We used the *yarn* package[33] to preprocess RNA-Seq data from GTEx release version 6.0[15] as *yarn* offers tissue-specific filtering and normalisation. This tissue-aware preprocessing ensures we do not filter out genes that show highly specific tissue expression. We specified the tissue groups listed in Table 1 of the publication by Paulson et al.[33] to *yarn*, as well as Bladder, Cervix Uteri and Fallopian Tube (these three tissues are not listed in Table 1 because the authors only used tissues with at least 15 samples). Genes that were not expressed (<1 count per million) in at least six samples (smallest tissue group size) were excluded. Tissue-aware normalisation was performed using the normalizeTissueAware() function and the smooth quantile normalisation (qsmooth) method[34]. After normalisation we averaged gene expression within each tissue group. The final expression table contains 31,542 genes and 41 tissues (Supplementary Data 6).

We selected human primary cell samples from the FANTOM5 data set[14] (excluding samples with poor RNA quality, shallow library depth or suspected mislabelling). Gene expression data (read counts of robust phase 1 CAGE peaks for human samples with annotation [hg19]) was downloaded using the FANTOM5 table extraction tool (TET). CAGE peaks with gene annotation (for a single gene) were retained and we summed peak counts within the same gene to obtain an overall measure of gene expression. We annotated the cell types and subtypes using the FANTOM5 ontology terms list (fantom.gsc.riken.jp/5/sstar/ FF_Ontology_terms_list). The plotCMDS() function from the *yarn* package was used to calculate multidimensional scaling (MDS) coordinates for the samples. Similarly to Paulson et al.[33], we iteratively examined the MDS plots to determine whether particular cell subtypes could be grouped together, resulting in a final set of 81 cell groups (Supplementary Data 7). We filtered and normalised the gene expression data by these cell groups (Normalisation_Group column in Supplementary Data 7), and removed genes not present in at least two samples due to the smaller group sizes in FANTOM5. After normalisation, we averaged gene expression by the 157 cell subtypes listed in Supplementary Data 7 (Cell_Type column). The final expression table contains 17,510 genes and 157 cell types (Supplementary Data 8).

**Specificity of gene expression**. To measure tissue or cell-specific gene expression we transformed both GTEx and FANTOM5 expression data sets to nonparametric specificity percentile scores as described by Hu et al.[35]. This transformation was applied to the expression data sets described above, after filtering out non-protein-coding genes. Firstly, nonparametric specificity scores are calculated by dividing the expression values for each gene by the sum of the squared expression values across that gene (Euclidean norm). Then, nonparametric specificity percentile scores were calculated by ranking the nonparametric specificity score for each tissue or cell type and dividing these ranks by the number of genes. Low scoring genes (close to zero) are specifically expressed, whereas high scoring genes (close to one) are either ubiquitously expressed or not expressed. The final expression table for GTEx

contains 18,442 genes and 41 tissues, and for FANTOM5 there are 16,096 genes and 157 cell types (Supplementary Data 9 and 10).

**VARiant Prioritisation by Phenotype (VARPP).** VARPP uses random forests[36] to predict pathogenicity of variants by phenotype whilst accounting for clustering of variants within genes. We used gene expression (GTEx or FANTOM5) in combination with variant prioritisation scores (CADD or MetaSVM) to predict pathogenicity. We chose CADD because it is widely used for variant filtering, and MetaSVM because we previously found it to be a top performer for variant prioritisation[13]. We also considered specificity of gene expression, because disease genes usually display tissue-specific expression[37]. Data and scripts required to run VAARP are available in the following GitHub repository https://github.com/deniando/VARPP along with instructions for use.

An overview of VARPP is shown in Supplementary Fig. 14. Given that gene expression (or specificity) is used as a predictor variable, multiple variants in the same gene will be assigned the same value. The default random forest algorithm assumes that observations are independent and hence does not take this clustering into account. When sampling for each tree of the random forest, a bootstrap sample (with replacement) is selected where approximately 63% of observations will be placed in-bag, with the remaining 37% placed out-of-bag[38]. The in-bag samples are used to build the tree, and the out-of-bag samples are used to internally validate the tree. This internal validation will be biased if the independence assumption is violated. In our case, sampling variants would result in overly optimistic performance of the classifier because variants in the same gene could be present in both in-bag and out-of-bag samples of a single tree. To mitigate this bias, we implemented a two-stage bootstrap approach for the random forests to account for clustering of variants within genes[39]. This involves bootstrap sampling with replacement at the gene level (rather than at the variant level), followed by selection of a single variant within each gene. Sampling at the gene level ensures that variants in the same gene can never be present in both in-bag and out-of-bag samples. Furthermore, we selected a single variant within each gene, rather than all variants, as this sampling method was shown to improve the performance of random forest[40].

Eight different classifiers were considered that differed in the predictor variables used, as follows:

- CADD + GTEx expression
- MetaSVM + GTEx expression
- CADD + FANTOM5 expression
- MetaSVM + FANTOM5 expression
- CADD + GTEx specificity percentile
- MetaSVM + GTEx specificity percentile
- CADD + FANTOM5 specificity percentile
- MetaSVM + FANTOM5 specificity percentile

We wrote a custom R function to implement the above, using the ranger package[41] to grow 2000 trees and to calculate unscaled permutation importance measures for each predictor variable. The permutation importance measure is calculated when predicting from each out-of-bag tree. Each predictor variable is randomly permuted in turn and change in accuracy of the predictions is calculated. Important variables will induce large changes in accuracy. Ranger does not handle missing values in predictor variables, hence we remove variants that do not have complete data across all predictor variables prior to growing each tree.

**Performance evaluation of VARPP.** We compared performance of VARPP versus using either CADD or MetaSVM alone to identify pathogenic variants. The R precrec package[42] was used to calculate the area under the precision-recall curve (auPRC) based on the interpolation method of Davis & Goadrich[43]. We chose to focus on the auPRC rather than the area under the receiver operating characteristic curve (auROC), due to the inherently imbalanced ratio of pathogenic to benign variants. In addition to the auPRC, we also calculated the proportion of true pathogenic variants (i.e. ClinVar pathogenic variants) in the top 100 predictions of pathogenicity (referred to as PP100 from this point on). This measure focuses on the top ranked variants as these are more likely to be considered for follow-up. In contrast, auPRC is an overall measure of the ranking of all variants, including the benign variants. We used a paired $t$-test to compare these two performance measures across all 1879 HPO terms, for VARPP versus the variant prioritisation tools.

**Simulated disease exomes.** To assess how well VARPP ranks a single causal variant within an individual exome we randomly spiked ClinVar pathogenic variants into exomes of healthy individuals obtained by the 1000 Genomes Project (1000 G). Variant call format (VCF) files for each chromosome were downloaded from http://ftp.1000genomes.ebi.ac.uk/vol1/ftp/release/20130502/. VCFtools[44] was used to remove variants with a minor allele frequency greater than 0.01. BCFtools (www.samtools.github.io/bcftools/) was used to extract each of the 2504 individuals from the multi-sample VCF files, whilst simultaneously removing rows that did not contain variants (i.e. rows where the individual was homozygous for the reference allele). The vcf-concat Perl script from VCFtools was used to concatenate the VCF files for each chromosome into one file per individual. We annotated variants in

each VCF file using dbNSFP version 2.9.3 (release March 12 2017) and based on this annotation we removed variants with minor allele frequencies greater than 0.01 in any of the following cohorts: (1) 1000 Genomes, (2) NHLBI GO Exome Sequencing Project, (3) Atherosclerosis Risk in Communities Study[45], and (4) Exome Aggregation Consortium.

We downloaded all ClinVar variants from ftp://ftp.ncbi.nlm.nih.gov/pub/clinvar/tab_delimited/ on 7 August 2018. Single nucleotide variants were filtered to those reported on build hg19 as this matches the build of the 1000 G VCF files. We selected variants assigned Clinvar pathogenic clinical significance, and also required variants to be annotated with one or more HPO terms. After filtering, 2365 ClinVar pathogenic variants remained as our spike-in variant set. The spike-in variant set contains 567 unique genes and 589 unique HPO annotations. We annotated our spike-in variant set with dbNSFP and annotation was returned for 2237 variants. We removed a further 8 variants with minor allele frequencies greater than 0.01 in any of the aforementioned cohorts (Supplementary Data 11).

A single spike-in variant was randomly allocated to the filtered set of rare variants for each 1000 G sample. Phenolyzer was queried using the HPO term(s) associated with each spike-in variant, and the corresponding 1000 G sample was retained for further analysis with VARPP if at least 25 seed genes were returned. Pathogenic variants within the seed genes were selected and VARPP was fitted as described above and tested back on each 1000 G sample. When testing back on each sample, the test set consisted of variants in genes that were not seen by VARPP. We wrote a custom R function to implement these steps. The whole process was carried out three times, so we could check for consistency across results (Supplementary Data 12). Performance was assessed by comparing ranks of the spike-in variant based on VARPP against ranks based on CADD or MetaSVM alone.

**Reporting summary.** Further information on research design is available in the Nature Research Reporting Summary linked to this article.

## Data availability
All data generated or analysed during this study are included in this published article (and its Supplementary Information files). The source data underlying Figs. 2 and 3 and Supplementary Figs. 1, 2, 3, 4, 5, 6, 7, and 8 are provided as Supplementary Data 1. The source data underlying Supplementary Figs. 9, 10, 11 and 12 are provided as Supplementary Data 2. The source data underlying Supplementary Fig. 13 are provided as Supplementary Data 3. A description of the Supplementary Data Files is available at https://doi.org/10.6084/m9.figshare.9808472. Supplementary Data Files 1, 2, 3, 4, 5, 6, 7, 8, 9, 10, 11 and 12 are available at https://doi.org/10.6084/m9.figshare.10010474, https://doi.org/10.6084/m9.figshare.10010480, https://doi.org/10.6084/m9.figshare.10010489, https://doi.org/10.6084/m9.figshare.9808445, https://doi.org/10.6084/m9.figshare.9808451, https://doi.org/10.6084/m9.figshare.9808454, https://doi.org/10.6084/m9.figshare.10010492, https://doi.org/10.6084/m9.figshare.9808457, https://doi.org/10.6084/m9.figshare.9808460, https://doi.org/10.6084/m9.figshare.9808463, https://doi.org/10.6084/m9.figshare.9808466 and https://doi.org/10.6084/m9.figshare.9808469, respectively.

## Code availability
Data and scripts required to run VAARP are available in the following GitHub repository https://github.com/deniando/VARPP along with instructions for use.

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

## Author contributions

D.A. performed analysis, interpreted results and drafted the manuscript. G.B. and J.M.B interpreted results and reviewed the manuscript. T.L. conceived the study, interpreted results and drafted the manuscript.

## Competing interests

The authors declare no competing interests.
