## [Peer Review File · Nature Communications]

Reviewers' comments:

Reviewer #1 (Remarks to the Author):

Anderson et al. presented a new approach to prioritizing the variants observed in Mendelian disease studies for the true causal variant(s). This work is an extension of their previous work published in *npj Genome Medicine*. The basic idea is that existing computational prediction of the pathogenicity of a variant is not phenotype-aware, so integrating phenotype related information can help to prioritizing the variants. The current practice of prioritizing typically involves multiple steps, one step identifies pathogenic variants of the gene regardless of the relativeness of the gene with the phenotype, and another step prioritizing the genes based on our knowledge of their relativeness with the phenotype. The authors want to merge those steps into one step. This idea is not entirely new. For example, the tool iCAGES (<https://doi.org/10.1186/s13073-016-0390-0>) also used multiple variant pathogenicity prediction scores and a phenotype relativeness score (using Phenolyzer, the same tool used by the authors) to produce an overall prioritizing score for the variant specific for cancer. The novelty of this study is that it is not specific to one disease but for all HPO terms, and it uses gene expression information.

One of my question is that it seems the authors are using gene expression information as a measure of relativeness of the gene to the phenotype. But the Phenolyzer has already provide a relativeness score, why the authors did not use this measure directly?

Another major concern is that many of the important information are missing from the method part so that I cannot fully judge the soundness of their approach. Especially, what are the training data for their random forest classifier and what are the testing data? What exactly are the features used in their classifier? I strongly encourage the authors to provide their training and testing data as supplementary material so that the reproducibility of the study can be increased.

Another concern is their simulated disease exomes. They spiked in pathogenic variants from clinvar to 1000G genomes and tested their new method on it to see whether the spiked in variants ranked high in the candidate list produced. My concern is that if the spiked in variants has already included when training the model, this test can be biased.

Some minor questions:

1. "We wrote a custom R function to implement the above, using the ranger package²⁶ to grow 2,000 trees and to calculate unscaled permutation importance measures for each predictor variable". How the importance measures are used in the VARPP method?
2. "We only considered gene panels, not extended gene panels". I know gene panels and extended gene panels are used in their *npj Genome Medicine* paper, but they shall briefly describe how those panels are produce. I also encourage the authors to provide the panels to increase the reproducibility of the study.
3. "After merging pathogenic variants into the gene panels for each HPO term". How are they merged?
4. "After filtering, there were 1,879 HPO terms containing 22,746 unique pathogenic variants across the gene panels". How the pathogenic variants were determined?
5. Supplementary tables are not available for review.

Reviewer #2 (Remarks to the Author):

General comments:

The authors propose to extend existing in-silico variant prioritisation tools by incorporating tissue-specific gene expression data into disease group specific classifiers. Improving the prediction performance of such tools is important to accurately interpret genetic variants identified in whole genome or exome sequencing data of patients. However, I have several serious concerns regarding data analysis, implementation and description as well as presentation of results.

Major comments:

Methods section:

To make the description of the approach easier to understand for the reader, I suggest several improvements:

- First describe the data and then the VARPP approach, followed by the criteria for the performance evaluation.
- I would use other headings for the subsections of the data section (e.g. pathogenic variants, benign variants, gene expression data).
- Add a figure that graphically shows how a disease specific classifier is trained (including genes, variants and predictor and outcome variables)

Selection of benign variants:

For a variant to be denoted as benign it had to have a MAF ≥ 0.01 in at least one population. I agree that this is a common approach to define benign variants but the authors should discuss if these MAF differences between pathogenic and benign variants might lead to overestimates of the prediction performance.

Random forest analysis:

More details are needed about the specifics of the random forests analyses:

- How were missing CADD or MetaSVM scores handled? The ranger implementation currently does not accept missing values.
- How was the imbalance dealt with? The ranger implementation offers several options, e.g. using weighting of observations (parameter `case.weights`) or classes (parameter `class.weights`). An alternative is to directly specify observations used for each tree using the parameter `inbag`.
- The authors should mention that random forests are not run in classification but in regression mode, meaning for each variant the probability of being pathogenic is estimated (probability machine as in Malley et al. 2011 *Methods of Information in Medicine* 51:74-81).

Variable importance:

- The class imbalance might also lead to wrong estimates of the variable importance since the prediction performance is based on overall accuracy. An alternative is to use the recently proposed corrected Gini importance (Nembrini, König and Wright 2018 *Bioinformatics* 34:3711-3718) which is implemented in the ranger function.
- Usually predictor variables are ranked depending on their estimated variable importance measurements since the absolute values cannot be interpreted sensibly. However, several variable selection methods exist that provide information which variables are important and which not (see Degenhardt, Seifert and Szymczak 2019 *Briefings in Bioinformatics* 20:492-503 for a recent review). The `vita` approach is implemented in the ranger package in the `importance_pvalues` function.
- The estimated prediction importance values are not correct since they are based on single trees (see software).

Performance evaluation:

- The authors used a paired t-test for comparing the performances between different approaches. However, the assumption of independent observations, i.e. variants, might not be fulfilled, in particular because disease relevant genes are selected for each classifier. The authors should at least add a remark that confidence intervals might be too small since the variance is underestimated.
- In addition to auPRC, the authors report ranking results based on the top 100 variants. How do results change if a lower or higher threshold than 100 is used?

Results section (Performance evaluation):

In the current manuscript the results are based on performance differences only. However, for a detailed evaluation, absolute performance values need to be specified. There is more room for improvement if the prediction based on the CADD or MetaSVM score alone is low compared to a well performing score. Furthermore, many comparisons stated in the text cannot be made based on the differences presented in table 1.

Software implementation:

I successfully installed and tested the software provided on GitHub. However, in the current form, for predicting the pathogenicity of some mutations, the random forest model needs to be trained each time which is time consuming and inconvenient. A more user-friendly implementation would provide separate functions for training and predicting new variants, e.g. by storing the random forest as an R object.

The documentation might also be improved to clearly state that the expression data and information about the benign variants are available and can be loaded with `"source(file="scripts/loadData.R")"`. In contrast, disease specific genes and variants need to be extracted from Phenolyzer and dbNSFP each time a new disease is considered.

I have some more questions:

- Why is ranger called separately for each tree? If ranger is called once for each model, the software can make use of the parallelization features of the implementation which will improve run time if many threads are available.
- Furthermore, estimating prediction importance based on a single tree does not work since it is based on the differences in prediction performance of the whole forest.
- Why does `VARPP_out$accuracy` reports only 61734 variants and not `nrow(disease_variants) + nrow(benign_variants) = 67078`? Also add the reason to the documentation.

Minor comments:

Methods section (Data subsection):

When describing the Phenolyzer approach the term gene panel might be confusing since it also refers to disease specific gene panels used for genetic testing. A better approach would be to use the Phenolyzer terminology of seed genes and addon genes.

Results section:

The first subsection is not really a result but rather a motivation of the approach. It might be better to move it to the introduction.

Table 2:

What is the meaning of the red color? This table might also be moved to the supplement.

Code and Software Submission Checklist:

Typical install time on a "normal" desktop computer is missing

Reviewer #1 (Remarks to the Author):

Anderson et al. presented a new approach to prioritizing the variants observed in Mendelian disease studies for the true causal variant(s). This work is an extension of their previous work published in *npj Genome Medicine*. The basic idea is that existing computational prediction of the pathogenicity of a variant is not phenotype-aware, so integrating phenotype related information can help to prioritizing the variants. The current practice of prioritizing typically involves multiple steps, one step identifies pathogenic variants of the gene regardless of the relativeness of the gene with the phenotype, and another step prioritizing the genes based on our knowledge of their relativeness with the phenotype. The authors want to merge those steps into one step. This idea is not entirely new. For example, the tool iCAGES (<https://doi.org/10.1186/s13073-016-0390-0>) also used multiple variant pathogenicity prediction scores and a phenotype relativeness score (using Phenolyzer, the same tool used by the authors) to produce an overall prioritizing score for the variant specific for cancer. The novelty of this study is that it is not specific to one disease but for all HPO terms, and it uses gene expression information.

Though iCAGES uses both multiple variant pathogenicity prediction scores and Phenolyzer scores they are used differently to how we have used them for VARPP. The major difference is that VARPP is not a phenotype aware algorithm, but instead produces classifiers for each phenotype or combination of phenotypes.

We are using Phenolyzer in a fundamentally different way compared to iCAGES. We use Phenolyzer to create training datasets, link those with expression and then apply phenotype-specific random forest classifiers. Unlike VARPP, iCAGES is not using Phenolyzer to select genes to use in the logistic regression models for driver gene prioritisation. Genes for the logistic regression models are selected based on prior knowledge (i.e. whether they are deemed as being significantly mutated genes in the TCGA Pan-Cancer cohort, or they are catalogued in the Cancer Gene Census).

One of my question is that it seems the authors are using gene expression information as a measure of relativeness of the gene to the phenotype. But the Phenolyzer has already provide a relativeness score, why the authors did not use this measure directly?

Our reason for using gene expression, is not solely as a measure of relativeness of the gene to the phenotype, but also to uncover associations between disease phenotype and biological function. We use cell and tissue-specific gene expression datasets to elucidate biological function because disease genes usually display tissue and/or cell-specific expression. This is a major advantage of VARPP, where in addition to better variant classification, we can also detect important links between disease phenotype and affected tissue and/or cell type.

The Phenolyzer confidence score represents the strength of association between the gene and the phenotype(s) based on the evidence in gene-disease databases. Given that we are using Phenolyzer to select disease genes, we were not able to include the confidence score in our classifiers as it would be self-selected as the best predictor (i.e. pathogenic variants within Phenolyzer disease genes will always have higher Phenolyzer confidence scores than benign variants which would arbitrarily have to be scored zero).

Another major concern is that many of the important information are missing from the method part so that I cannot fully judge the soundness of their approach. Especially, what are the training data for their random forest classifier and what are the testing data? What exactly are the features used in their classifier? I strongly encourage the authors to provide their training and testing data as supplementary material so that the reproducibility of the study can be increased.

We apologise that the methodology was not clear. We have added an overview of VARPP (Supplementary Figure 14 and below) to address this key issue. To clarify, random forests do not use a separate set of training and testing data because the data are split into training (in-bag-sample for growing tree) and validation (out-of-bag sample for validating tree) sets during growth of each tree in the random forest (see visual overview). Use of out-of-bag samples for internal validation of each tree ensures that random forests return unbiased measures of accuracy. The features used by VARPP include CADD or MetaSVM scores and either cell or tissue-specific gene expression (this information has been added to the visual overview). We have now provided the full set of pathogenic variants (across the 1,879 Human Phenotype Ontology terms) and the benign variants as supplementary data (Supplementary Data 1 & 2 respectively) for reproducibility of our results. In addition, the software to reproduce our results is available on GitHub (<https://github.com/deniando/VARPP>).

Gene 1	v1	v2	v3	v4					
Gene 2	v1	v2	v3	v4	v5	v6	v7		
Gene 3	v1								
Gene 4	v1	v2	v3	v4	v5				
Gene 5	v1	v2	v3						
Gene 6	v1	v2	v3	v4	v5	v6	v7		
Gene 7	v1	v2	v3	v4	v5	v6	v7		
Gene 8	v1	v2	v3	v4	v5	v6	v7		
Gene 9	v1	v2	v3	v4	v5	v6	v7		
Gene 10	v1	v2	v3	v4	v5	v6	v7		

Red = ClinVar pathogenic variants within genes associated with HPO phenotypic abnormality term(s)

Blue = Benign variants

* Benign variants are selected from genes that have no reported ClinVar variants, hence these genes do not overlap the genes containing ClinVar pathogenic variants.

Features associated with each variant are used by the classifier and include CADD or MetaSVM scores in conjunction with either cell or tissue-specific gene expression.

Sample genes with replacement, followed by selection of a single variant for in-bag sample (the same variant is selected for repeatedly sampled genes). Genes not selected will be the out-of-bag sample.

In-bag sample

Gene 1	v2
Gene 4	v5
Gene 4	v5
Gene 4	v5
Gene 4	v5
Gene 5	v1
Gene 6	v3
Gene 8	v7
Gene 9	v1
Gene 10	v3

Grow tree using in-bag sample

At each node \bigcirc , a subset ($\sqrt{\#}$ features) of the features are randomly chosen and the best splitter is selected

Pipe out-of-bag sample down tree

Tree casts a vote where 1=Pathogenic and 0=Benign

	Out-of-bag sample							Tree vote						
Gene 2	v1	v2	v3	v4	v5	v6	v7	1	1	0	1	0	1	1
Gene 3	v1							1						
Gene 7	v1	v2	v3	v4	v5	v6	v7	0	0	1	0	0	1	0

Tree votes for out-of-bag samples are collected across all trees in the forest and the predicted probability of pathogenicity for each variant is calculated as the number of pathogenic predictions divided by the number of trees casting votes.

Another concern is their simulated disease exomes. They spiked in pathogenic variants from clinvar to 1000G genomes and tested their new method on it to see whether the spiked in variants ranked high

in the candidate list produced. My concern is that if the spiked in variants has already included when training the model, this test can be biased.

When growing individual trees in the random forest, spiked-in variants will either be in-bag or out-of-bag due to our sampling scheme (see above visual overview of VARPP). If the spiked in variant is selected in-bag, it will be used to grow the tree but a prediction of pathogenicity for the spike-in variant will not be made. If the spiked in variant is selected out-of-bag, a prediction of pathogenicity for the spike-in variant will be made but it will not have been used to grow the tree. Hence predictions of pathogenicity for the spike-in variant are not biased.

Some minor questions:

1. “We wrote a custom R function to implement the above, using the ranger package²⁶ to grow 2,000 trees and to calculate unscaled permutation importance measures for each predictor variable”. How the importance measures are used in the VARPP method?

The importance measures are not used directly by VARPP, rather they are returned by ranger. Our interest in the importance measures is to identify which cell or tissue-types show association with the Human Phenotype Ontology term(s) being considered, as the premise of our approach is that biologically relevant cell or tissue-types should prove useful when prioritising variants. The importance measures are returned to the user when they use VARPP, in order to reveal potentially interesting associations between cell and/or tissue type and disease.

2. “We only considered gene panels, not extended gene panels”. I know gene panels and extended gene panels are used in their npj Genome Medicine paper, but they shall briefly describe how those panels are produce. I also encourage the authors to provide the panels to increase the reproducibility of the study.

We have removed reference to extended gene panels, given that we do not use such panels in our study. We changed the terminology based on reviewer comments so that gene panels are now referred to as seed genes to be consistent with Phenolyzer. In Methods (Linking disease phenotypes to genes using Phenolyzer) the description is now worded as follows:

“We only considered genes with a known disease association (seed genes in Phenolyzer).”

We have also provided the gene panels as Supplementary Data 1 for reproducibility of our results.

3. “After merging pathogenic variants into the gene panels for each HPO term”. How are they merged?

Though we do merge the pathogenic variants into the seed genes (using the R `merge()` function), we reworded this sentence in Methods (Pathogenic variants) to be less technical as follows (Again, please note the change of terminology from gene panels to seed genes):

“After selecting pathogenic variants within the seed genes for each HPO term, we excluded terms having fewer than 25 genes.”

The R object storing these seed genes and associated pathogenic variants has now been made available as Supplementary Data 1.

4. “After filtering, there were 1,879 HPO terms containing 22,746 unique pathogenic variants across the gene panels”. How the pathogenic variants were determined?

Pathogenic variants were determined using ClinVar clinical significance (<https://www.ncbi.nlm.nih.gov/clinvar/docs/clinsig/>). ClinVar is a database that describes the pathogenicity of variants implicated in Mendelian disorders. ClinVar uses the five clinical significance categories recommended by the American College of Medical Genetics and Genomics (benign, likely benign, uncertain significance, likely pathogenic and pathogenic). We used the most stringent definition for pathogenicity. We amended the second sentence in Methods (Pathogenic variants) to make this clear:

“We selected ClinVar pathogenic variants and used the converted rank scores for CADD and MetaSVM.”

5. Supplementary tables are not available for review.

Our apologies for this oversight, Supplementary Table 1 was missing from the submission and has now been included (it is now Supplementary Table 5, rather than Supplementary Table 1).

Reviewer #2 (Remarks to the Author):

General comments:

The authors propose to extend existing in-silico variant prioritisation tools by incorporating tissue-specific gene expression data into disease group specific classifiers. Improving the prediction performance of such tools is important to accurately interpret genetic variants identified in whole genome or exome sequencing data of patients. However, I have several serious concerns regarding data analysis, implementation and description as well as presentation of results.

Major comments:

Methods section:

To make the description of the approach easier to understand for the reader, I suggest several improvements:

- First describe the data and then the VARPP approach, followed by the criteria for the performance evaluation.

Great suggestion, we have reordered the methods sections accordingly.

- I would use other headings for the subsections of the data section (e.g. pathogenic variants, benign variants, gene expression data).

We simplified the headings for the subsections of the data section as suggested.

- Add a figure that graphically shows how a disease specific classifier is trained (including genes, variants and predictor and outcome variables)

We have added the below figure as Supplementary Figure 14. Lack of such a figure was pointed out by another reviewer. The revised manuscript includes a new visual overview of VARPP to make the method clearer.

Gene 1	v1	v2	v3	v4			
Gene 2	v1	v2	v3	v4	v5	v6	v7
Gene 3	v1						
Gene 4	v1	v2	v3	v4	v5		
Gene 5	v1	v2	v3				
Gene 6	v1	v2	v3	v4	v5	v6	v7
Gene 7	v1	v2	v3	v4	v5	v6	v7
Gene 8	v1	v2	v3	v4	v5	v6	v7
Gene 9	v1	v2	v3	v4	v5	v6	v7
Gene 10	v1	v2	v3	v4	v5	v6	v7

Red = ClinVar pathogenic variants within genes associated with HPO phenotypic abnormality term(s)

Blue = Benign variants

* Benign variants are selected from genes that have no reported ClinVar variants, hence these genes do not overlap the genes containing ClinVar pathogenic variants.

Features associated with each variant are used by the classifier and include CADD or MetaSVM scores in conjunction with either cell or tissue-specific gene expression.

Sample genes with replacement, followed by selection of a single variant for in-bag sample (the same variant is selected for repeatedly sampled genes). Genes not selected will be the out-of-bag sample.

In-bag sample

Gene 1	v2
Gene 4	v5
Gene 4	v5
Gene 4	v5
Gene 4	v5
Gene 5	v1
Gene 6	v3
Gene 8	v7
Gene 9	v1
Gene 10	v3

Grow tree using in-bag sample

At each node \bigcirc , a subset ($\sqrt{\# \text{ features}}$) of the features are randomly chosen and the best splitter is selected

Pipe out-of-bag sample down tree

Tree casts a vote where 1=Pathogenic and 0=Benign

	Out-of-bag sample							Tree vote							
	v1	v2	v3	v4	v5	v6	v7								
Gene 2	v1	v2	v3	v4	v5	v6	v7	1	1	0	1	0	1	1	
Gene 3	v1							1							
Gene 7	v1	v2	v3	v4	v5	v6	v7	0	0	1	0	0	1	0	

Tree votes for out-of-bag samples are collected across all trees in the forest and the predicted probability of pathogenicity for each variant is calculated as the number of pathogenic predictions divided by the number of trees casting votes.

Selection of benign variants:

For a variant to be denoted as benign it had to have a MAF ≥ 0.01 in at least one population. I agree that this is a common approach to define benign variants but the authors should discuss if these MAF

differences between pathogenic and benign variants might lead to overestimates of the prediction performance.

Any bias would be most prevalent for the MetaSVM and CADD scores which are at the variant level, rather than gene expression at the gene level (i.e. all genes likely contain variants with MAF ≥ 0.01 and hence will be available for selection). The below plot shows the normalised distribution of MetaSVM and CADD scores across dbNSFP variants that are also present in 1000 Genomes, ALSPAC or ExAC. Variants have been annotated based on minor allele frequencies (MAF) < 0.01 or ≥ 0.01 and whether they are known benign variants according to ClinVar ($n=5,622$). Distribution of MetaSVM and CADD scores across these categories is similar for all three cohorts. Our strategy of selecting variants with MAF ≥ 0.01 does select variants that are skewed toward lower MetaSVM and CADD scores, but this is to be expected given that we are targeting benign variants which will have lower MetaSVM and CADD scores.

ClinVar benign variants with MAF ≥ 0.01 have lower MetaSVM and CADD scores than ClinVar benign variants with MAF < 0.01 , which illustrates the decreasing accuracy of these tools in classifying variants as MAF decreases. In addition, when considering variants with MAF < 0.01 that are not in ClinVar, a large spike is evident for CADD scores close to 1, so CADD is predicting a high number of rare functional variants for MAF < 0.01 . We think this supports the strategy of selecting variants with MAF ≥ 0.01 as a more reliable set of benign variants. There is still noise in the MetaSVM and CADD scores when selecting variants with MAF ≥ 0.01 , as evidenced by many ClinVar benign variants with high scores, but there is clearly much more noise when considering variants with MAF < 0.01 .

Importantly though, if we plot the unnormalised distributions (below), it can be seen that even if we had included ClinVar benign variants with MAF < 0.01 , the effect on our estimates of performance would be negligible due to the small number of such variants. Nevertheless, we have added the following sentences to paragraph 3 of the Discussion:

“Further to the performance of VARPP, it should also be mentioned that our strategy of selecting variants with minor allele frequencies of at least 0.01 as our benign set of variants could bias performance estimates of VARPP, because benign variants with minor allele frequencies < 0.01 may

differ in their distribution of CADD and MetaSVM scores. Whether this could lead to over or underestimation of VARPPs performance is impossible to assess due to lack of reliable annotation for rare benign variants. Any bias would be minimal though, due to our large sample of benign variants, and our strategy is preferable to avoid inclusion of rare functional variants.”

Random forest analysis:

More details are needed about the specifics of the random forests analyses:

- How were missing CADD or MetaSVM scores handled? The ranger implementation currently does not accept missing values.

Prior to growing each tree with ranger, variants with missing values are dropped. This is done separately for CADD and MetaSVM because they have differing proportions of missing scores (as described in Methods: final sentence of Benign variants section). We decided not to restrict the variants to those where there was no missing data because we would lose potentially important variants that are important for prediction by phenotype. We have added the following sentence to the Methods (last sentence of VARiant Prioritisation by Phenotype (VARPP) section):

“Ranger does not handle missing values in predictor variables, hence we remove variants that do not have complete data across all predictor variables prior to growing each tree.”

We also added the following to paragraph 3 of the Discussion:

“Finally, the higher proportion of missing scores for MetaSVM (15.2% across all pathogenic and benign variants) in contrast to CADD (0.03%), means comparisons across VARPP classifiers is imperfect. This also needs to be considered when using VARPP to predict pathogenicity of an individual patient’s variants, as less will be scored for VARPP classifiers including MetaSVM compared to VARPP classifiers including CADD.”

- How was the imbalance dealt with? The ranger implementation offers several options, e.g. using weighting of observations (parameter case.weights) or classes (parameter class.weights). An alternative is to directly specify observations used for each tree using the parameter inbag.

The aim of VARPP is to find the needle in the haystack and our approach emulates this. We did consider downsampling the benign variants to match the number of pathogenic variants for each HPO term, but this resulted in poor performance of VARPP. This is because some HPO terms are associated with a small number of pathogenic variants (< 100) and selecting the same small number of benign variants introduced noise. We instead select from the same set of benign variants for every phenotype because this more closely resembles the variant filtering task performed on individual patients (i.e. better matching of training and test datasets). We chose to assess VARPP's performance using the area under the precision-recall curve because it is robust to imbalanced data. Random forest has also been shown to deal well with imbalanced data [D. J. Dittman, T. M. Khoshgoftaar and A. Napolitano, "The Effect of Data Sampling When Using Random Forest on Imbalanced Bioinformatics Data," *2015 IEEE International Conference on Information Reuse and Integration*, San Francisco, CA, 2015, pp. 457-463. Doi: 10.1109/IRI.2015.76].

- The authors should mention that random forests are not run in classification but in regression mode, meaning for each variant the probability of being pathogenic is estimated (probability machine as in Malley et al. 2011 *Methods of Information in Medicine* 51:74-81).

VARRP is run in classification mode, not regression mode. We hope the visual overview (Supplementary Figure 14) clarifies this point. We collect votes across individual trees and calculate the probability of pathogenicity for each variant as the number of pathogenic predictions divided by the number of trees casting votes.

Variable importance:

- The class imbalance might also lead to wrong estimates of the variable importance since the prediction performance is based on overall accuracy. An alternative is to use the recently proposed corrected Gini importance (Nembrini, König and Wright 2018 *Bioinformatics* 34:3711-3718) which is implemented in the ranger function.

Yes, we agree that the variable importance used by VARPP is not ideal and we outlined the shortcomings in paragraph 3 of the discussion. Nevertheless, we believe this measure has utility in relative terms when used to compare importance amongst predictor variables. We chose the permutation importance over the Gini impurity importance because it is known that the Gini impurity importance is biased towards selecting variables with the most split points [Strobl, C, Boulesteix, AL, Zeileis, A, Hothorn, T (2007). Bias in random forest variable importance measures: illustrations, sources and a solution. *BMC Bioinformatics*, 8:25.]. We note that the corrected Gini impurity performance also addresses this problem, but also note that it has only been assessed on datasets with minor imbalance. Therefore, it is unclear whether there is a major advantage over the permutation importance in our case. Evaluating this as an alternative to the permutation importance is beyond the scope of this manuscript, but we will implement this in the future to improve the running times of VARPP.

In summary, we agree that investigating different variable importance measures represents a fruitful future direction but wish to point the reviewer to Supplementary Table 3, highlighting that the current variable importance measure can reveal biologically plausible disease/tissue associations.

- Usually predictor variables are ranked depending on their estimated variable importance measurements since the absolute values cannot be interpreted sensibly. However, several variable selection methods exist that provide information which variables are important and which not (see Degenhardt, Seifert and Szymczak 2019 Briefings in Bioinformatics 20:492-503 for a recent review). The vita approach is implemented in the ranger package in the importance_pvalues function.

We appreciate your suggestion and note that Vita is highly recommended by the authors of the referenced study. As per the corrected Gini importance measure above, evaluating this is beyond the scope of this manuscript, but we will offer this in a future version of VARPP. And again we would like to point out that Supplementary Table 3 does highlight biologically plausible disease/tissue associations.

- The estimated prediction importance values are not correct since they are based on single trees (see software).

The final importance values returned by VARPP are not based on single trees. As mentioned above, we grow random forests manually with VARPP (i.e. a tree at a time) because we were not able to implement our two-stage sampling scheme directly with ranger. As per the visual overview (Supplementary Figure 14) where we collect tree votes for each tree, we also collect variable importances for each tree. The variable importances are then averaged across all trees the predictor is present in (as described by Breiman 2001) and this average is reported by VARPP.

Performance evaluation:

- The authors used a paired t-test for comparing the performances between different approaches. However, the assumption of independent observations, i.e. variants, might not be fulfilled, in particular because disease relevant genes are selected for each classifier. The authors should at least add a remark that confidence intervals might be too small since the variance is underestimated.

We have added the following sentence to paragraph 3 of the Discussion:

“Furthermore, the confidence intervals presented in Table 1 may be too narrow given the overlap of genes and variants across HPO terms.”

- In addition to auPRC, the authors report ranking results based on the top 100 variants. How do results change if a lower or higher threshold than 100 is used?

The threshold of 100 was chosen because it is close to the minimum number of pathogenic variants associated with some HPO terms. This threshold is also sensible from a practical standpoint, as users of VARPP are unlikely to consider more than the top 100 predictions of pathogenicity. The below plot shows the results as per Figure 2 of the manuscript, if we increase the threshold to 200. In this case 64% of terms improve for VARPP including CADD and GTEX, and 78% of terms improve for VARPP including MetaSVM and GTEX.

The below plot shows the results as per Figure 2 of the manuscript, if we decrease the threshold to 50. In this case 41% of terms improve for VARPP including CADD and GTEEx, and 89% of terms improve for VARPP including MetaSVM and GTEEx. We do not claim that VARPP will improve on predictions of variant pathogenicity for all HPO terms (or combinations of HPO terms), but the relevant performance statistics are returned to the user and this allows them to decide whether VARPP scores are preferred over use of CADD or MetaSVM scores alone. We also imagine users may use VARPP scores in conjunction with CADD or MetaSVM scores as per the recommendation of the American College of Medical Genetics (i.e. consultation of more than one variant prioritisation tool is recommended when prioritising variants based on in silico pathogenicity predictions).

Results section (Performance evaluation):

In the current manuscript the results are based on performance differences only. However, for a detailed evaluation, absolute performance values need to be specified. There is more room for improvement if the prediction based on the CADD or MetaSVM score alone is low compared to a well performing score. Furthermore, many comparisons stated in the text cannot be made based on the differences presented in table 1.

We have added Supplementary Table 1 which lists the absolute performance values. We reworded the comparisons stated in the text to more accurately reflect that improvements in performance are seen for some, not all terms.

Software implementation:

I successfully installed and tested the software provided on GitHub. However, in the current form, for predicting the pathogenicity of some mutations, the random forest model needs to be trained each time which is time consuming and inconvenient. A more user-friendly implementation would provide separate functions for training and predicting new variants, e.g. by storing the random forest as an R object.

Based on our use of VARPP to prioritise variants in individual patient exomes, we do not anticipate the need to store a random forest for further prediction. Our group has a rare diseases program where we attempt to find causal variants related to a patient's phenotype(s). Clinicians assign many HPO terms to describe the patient's phenotype and we have found that it is rare to find two patients with the same set of HPO terms. In practice we would only run VARPP once on the applicable set of HPO terms for each patient. The VARPP scores are then collated into a report alongside other variant annotation. We are considering implementing VARRP in C in the future to reduce running times since we want to run it each time we get new patient data.

The documentation might also be improved to clearly state that the expression data and information about the benign variants are available and can be loaded with "source(file="scripts/loadData.R")". In contrast, disease specific genes and variants need to be extracted from Phenolyzer and dbNSFP each time a new disease is considered.

We have added additional information to the GitHub repository (<https://github.com/deniando/VARPP>) to better explain this under "Instructions for Use".

I have some more questions:

- Why is ranger called separately for each tree? If ranger is called once for each model, the software can make use of the parallelization features of the implementation which will improve run time if many threads are available.

As mentioned above, we grow random forests manually with VARPP (i.e. a tree at a time) because we were not able to implement our two-stage sampling scheme directly with ranger. We also seek improvements in run time, and to this end we plan to rewrite VARPP in C.

- Furthermore, estimating prediction importance based on a single tree does not work since it is based on the differences in prediction performance of the whole forest.

As mentioned above, votes and variable importances are stored as we grow each tree, but the final calculations of predicted pathogenicity and variable importance are based on all trees in the forest (as described by Breiman 2001). Hence, none of our predictions of pathogenicity or variable importances are based on a single tree.

- Why does VARPP_out\$accuracy reports only 61734 variants and not nrow(disease_variants) + nrow(benign_variants) = 67078? Also add the reason to the documentation.

This is because we remove variants that were not seen by VARPP prior to calculating predicted probabilities of pathogenicity. The reason a variant would not be seen by VARPP is because of missing data in either MetaSVM or CADD scores, or in gene expression. If a particular variant does not have complete data across all of these predictors, then a prediction of pathogenicity cannot be made by VARPP.

This also applies to predictions of pathogenicity made for individual patients (i.e. variants present in the user specified vcf file), but in this case we do not drop variants that were not seen by VARPP, rather they are returned to the user with a predicted probability of NA. The GitHub page (<https://github.com/deniando/VARPP>) does explain this when describing the VARPP output for the patient variants (“Some variants will not have predictions due to missing data in either CADD/MetaSVM scores or gene expression”). We have added the same sentence to the GitHub page when describing the accuracy output.

Minor comments:

Methods section (Data subsection):

When describing the Phenolyzer approach the term gene panel might be confusing since it also refers to disease specific gene panels used for genetic testing. A better approach would be to use the Phenolyzer terminology of seed genes and addon genes.

We have amended the terminology as suggested. We have also removed reference to extended gene panels (addon genes) given that they are not considered in this study.

Results section:

The first subsection is not really a result but rather a motivation of the approach. It might be better to move it to the introduction.

We have moved this subsection to the introduction as suggested.

Table 2:

What is the meaning of the red color? This table might also be moved to the supplement.

Our apologies for not explaining the meaning of the red colour. This was used to highlight biologically plausible associations, but this should have been removed prior to submission. This table has been moved to Supplementary Table 3.

Code and Software Submission Checklist:

Typical install time on a "normal" desktop computer is missing

We have added the following sentence to the GitHub page for VARPP under "Software Requirements":

"The installation time on a standard computer with the aforementioned specifications is approximately 45 minutes."

REVIEWERS' COMMENTS:

Reviewer #1 (Remarks to the Author):

The authors have answered my questions.

Reviewer #2 (Remarks to the Author):

The authors have addressed most of my concerns and suggestions in the revised version of the manuscript. Especially the new supplementary figure 14 showing the different steps of training the classifier is really helpful for better understanding of the underlying method.

However, I have some comments regarding the ranking results and the implementation:

1. Ranking results based on the top 100 variants

In their response letter, the authors present some additional results based on the top 200 and top 50 variants. This information should be added to the manuscript, i.e. that qualitative results (larger number of improvements for MetaSVM) do not change if the top 50 or top 200 variants are used instead of top 100.

And explain why the top 50 and not top 100 are used for the analysis of the simulated disease exomes?

2. Implementation

As the authors acknowledge, the current implementation is slow because of calling the ranger function separately for each tree. However, since version 0.11.0 it is now possible to manually select observations for each tree and provide this information to the ranger function using the `inbag` argument. Thus, ranger could be used very efficiently both regarding run time and memory consumption.

The authors should add a paragraph to the discussion describing how they plan to improve the implementation in the future (e.g. alternate variable importance measure such as the corrected Gini importance or variable selection). I fully understand that for their specific application scenario of single patients with specific symptoms it might not be necessary to separate training of the classifier from using it for predicting pathogenicity. However, to enable a broader applicability in the context of Mendelian or also complex diseases, this feature would be really helpful.

In the documentation it should be stated explicitly that accuracy contains only variants for which both CADD and MetaSVM scores were available and that `patient_predictions` contain NA for those variants.

I also noted that the function assumes that the object `benign_variants` exists in the current environment. To make the code more robust, it should be added as an argument to the VARPP function.

Reviewer #1 (Remarks to the Author):

The authors have answered my questions.

Reviewer #2 (Remarks to the Author):

The authors have addressed most of my concerns and suggestions in the revised version of the manuscript. Especially the new supplementary figure 14 showing the different steps of training the classifier is really helpful for better understanding of the underlying method.

However, I have some comments regarding the ranking results and the implementation:

1. Ranking results based on the top 100 variants

In their response letter, the authors present some additional results based on the top 200 and top 50 variants. This information should be added to the manuscript, i.e. that qualitative results (larger number of improvements for MetaSVM) do not change if the top 50 or top 200 variants are used instead of top 100.

We have added the following sentences to the end of the third paragraph of Results (Performance of VARPP):

“We further note that improvements are consistent for VARPP including MetaSVM if instead we use the proportion of true pathogenic variants in the top 50 or 200 predictions of pathogenicity (PP50=1,664 [89%] of HPO terms improve; PP200=1,467 [78%] of HPO terms improve). This is also the case for VARPP including CADD for the PP200 (1,198 [64%] of HPO terms improve), but there are fewer improvements for the PP50 (776 [41%] of HPO terms improve) due to HPO terms where true pathogenic variants in the top 50 have CADD scores close to 1 (i.e. there is little opportunity for VARPP to improve on predictions of variant pathogenicity in these scenarios).”

And explain why the top 50 and not top 100 are used for the analysis of the simulated disease exomes?

We lowered the threshold here because the simulated disease exomes have a single pathogenic variant, whereas assessments of VARPPs performance across the 1,879 HPO terms have anywhere between 96 and 19,549 pathogenic variants (PP100 was chosen because it is close to the minimum of 96).

2. Implementation

As the authors acknowledge, the current implementation is slow because of calling the ranger function separately for each tree. However, since version 0.11.0 it is now possible to manually select observations for each tree and provide this information to the ranger function using the inbag argument. Thus, ranger could be used very efficiently both regarding run time and memory consumption.

Thank you for pointing this out as use of the `inbag` argument should greatly improve the run time of VARPP. When we performed the analyses for this manuscript we used ranger version 0.9.0 so we did not have access to the `inbag` argument. For our purposes, the `inbag` argument does not work “out of the box” due to issues noted here <https://github.com/imbs-hl/ranger/issues/142> regarding the need for a third “discard” category. However, we will code a workaround to get this running given the anticipated gain in run time. We will implement this change in the next version of VARPP, alongside offering the additional features outlined in the next paragraph.

The authors should add a paragraph to the discussion describing how they plan to improve the implementation in the future (e.g. alternate variable importance measure such as the corrected Gini importance or variable selection). I fully understand that for their specific application scenario of single patients with specific symptoms it might not be necessary to separate training of the classifier from using it for predicting pathogenicity. However, to enable a broader applicability in the context of Mendelian or also complex diseases, this feature would be really helpful.

We have added the following sentences to the end of the fifth paragraph of the discussion:

“We will improve implementation of VARPP including offering actual impurity reduction (AIR) variable importance²⁰ and variable importance p -values^{21,22}, as additional sources of evidence for determining variable importance. We are also currently working on improving run times for VARPP and on separate functions for training and prediction, for the convenience of users who would like to use the same random forest repeatedly for predictions on new data.”

In the documentation it should be stated explicitly that accuracy contains only variants for which both CADD and MetaSVM scores were available and that patient_predictions contain NA for those variants.

We have reworded this on the VARPP Github page (<https://github.com/deniando/VARPP>) under Instructions for Use:

- `accuracy`: Out of bag predicted probabilities of pathogenicity from VARPP. These are used to assess the performance of VARPP for the queried phenotype. There are two classifiers; `CADD_expression` CADD scores in combination with expression data and `MetaSVM_expression` MetaSVM scores in combination with expression data. In addition, `CADD_raw_rankscore` and `MetaSVM_rankscore` are the CADD and MetaSVM scores, whose performance can be compared to VARPP. Some variants will be omitted from this data frame due to missing data in either CADD/MetaSVM scores or gene expression.
- `patient_predictions`: Predicted probabilities of pathogenicity from VARPP for the patient variants. These predictions are in columns `CADD_expression` and `MetaSVM_expression`. The remaining columns are the dbNSFP annotations merged in from the `patient_variants` file. All patient variants will be returned but some variants will have NA predictions due to missing data in either CADD/MetaSVM scores or gene expression.

I also noted that the function assumes that the object `benign_variants` exists in the current environment. To make the code more robust, it should be added as an argument to the VARPP function.

We have implemented this change.